# TokMem: One-Token Procedural Memory for Large Language Models

**Zijun Wu**[1]**, Yongchang Hao**[1]**, Lili Mou**[1,2]
[1]Dept. Computing Science & Alberta Machine Intelligence Institute (Amii), University of Alberta
[2]Canada CIFAR AI Chair
{zijun4,yongcha1}@ualberta.ca, doublepower.mou@gmail.com

## Abstract

Large language models are typically controlled via prompts, which must be repeatedly re-processed for every new query and are difficult to reuse modularly. We introduce TokMem, a procedural memory framework that compiles each reusable task procedure into a single trainable memory token. Each token serves as both a procedure index and a generation control signal that steers generation, enabling targeted behaviors with constant-size overhead. TokMem keeps the backbone LLM frozen and stores procedural knowledge entirely in these dedicated units, so new procedures can be added continually without interfering with existing ones. We evaluate TokMem on two settings: atomic recall over 1,000 Super-Natural Instructions tasks and compositional recall on multi-step function-calling. Our results show that TokMem consistently outperforms retrieval-augmented prompting while avoiding repeated context overhead. Moreover, it matches or exceeds parameter-efficient fine-tuning with substantially fewer trainable parameters.[1]

## 1 Introduction

Large language models (LLMs) have become the foundation of modern natural language processing, powering a wide range of applications in text understanding, generation, and coding (Brown et al., 2020; Llama Team, 2024; Chen et al., 2021). Prompting is a widely adopted way to steer LLM behavior, where in-context learning enables adaptation to new tasks without parameter updates (Brown et al., 2020). Consequently, prompt and context engineering has emerged as a dominant interface for specifying tasks, obtaining relevant information, and guiding multi-step reasoning or tool invocation (Wei et al., 2022b; Yao et al., 2023; Sahoo et al., 2025).

Despite their effectiveness, long prompts are inherently inefficient. They are labor-intensive to construct and maintain, and they do not scale well when conditioning diverse queries across different tasks (Liu et al., 2023). At inference time, longer contexts increase compute because self-attention scales quadratically (Vaswani et al., 2017). They also shrink the usable context window for inputs and outputs, often causing truncation and information loss (Liu et al., 2024a). Moreover, as tasks accumulate, these costs make it increasingly difficult to execute task-specific behaviors efficiently.

To mitigate these issues, recent work externalizes prompts into retrieval-based memory. Retrieval-augmented generation (RAG; Lewis et al., 2020) is used to retrieves and reinserts documents or conversational state at inference time, a strategy exemplified by MemGPT (Packer et al., 2024). Although retrieving in-context demonstrations (Wei et al., 2022b) can supply procedural cues, the retrieved knowledge remains explicit text that must be repeatedly interpreted, resembling declarative memory in cognitive science. This leads to two drawbacks: (1) retrieved content still occupies the context window, reintroducing quadratic compute and truncation pressure, and (2) frequently used procedures are repeatedly re-read as text rather than distilled into compact, reusable representations.

We propose One-**Tok**en Procedural **Mem**ory (**TokMem**), a modular framework that encodes task procedures into compact, trainable tokens while maintaining a frozen LLM backbone. Here, a *procedure* refers to a reusable context–response mapping that captures a specific task behavior; the terminology is inspired by physiological research (Anderson & Lebiere, 1998), which posits

---

[1]Our code is publicly available at https://github.com/MANGA-UOFA/TokMem

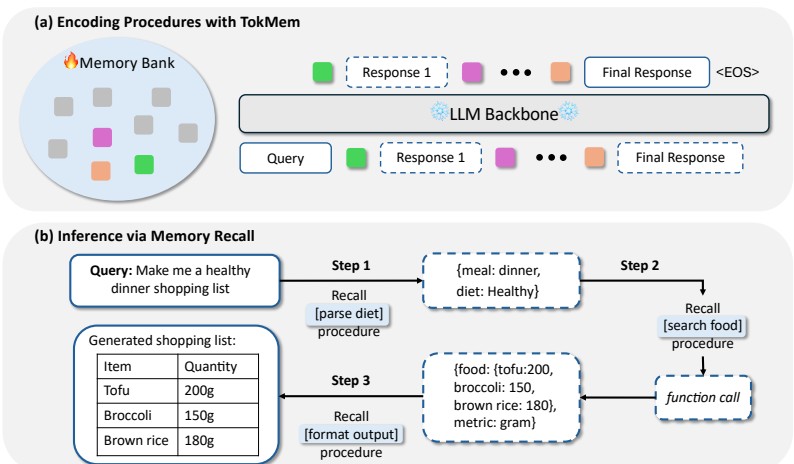

Figure 1: Overview of our TokMem approach. (a) New (colored) memory tokens are interleaved with text sequences and trained via next-token prediction while the LLM backbone remains frozen. (b) At inference time, a query recalls and chains memory tokens (e.g., parse, search, format), enabling multi-step procedural behavior without long prompts.

procedural knowledge as a compiled form of skill that is more efficient than the slow interpretation of facts. Unlike factual knowledge that can be expressed in plain text, procedural knowledge (e.g., riding a bicycle) is often more nuanced and sophisticated.

In TokMem, a memory bank stores learned procedures (Figure 1a), where each memory token serves both as an index to a procedure and as a control signal that steers generation, allowing the model to execute targeted tasks with constant-size overhead regardless of the procedure's complexity. Rather than front-loading procedures as long prompts, TokMem integrates memory tokens directly into the generation process by recalling them on demand at each stage. As illustrated in Figure 1b, TokMem supports multi-step workflows by sequentially recalls procedure tokens such as parsing, searching, and formatting; after the model completes a specific segment of the response, it retrieves the next relevant token to condition the following stage, allowing for modular task composition.

A key advantage of TokMem is that its memory tokens are parameter-isolated from the backbone LLM. Because each procedure is distilled into its own dedicated token, new skills can be added without interfering with or degrading existing ones. This isolation naturally supports continual learning, enabling the model to accumulate procedural skills over time while remaining stable. This mirrors human procedural memory, where skills are gradually acquired through practice and later invoked by contextual cues (Anderson & Lebiere, 1998). Overall, TokMem enables efficient learning and continual expansion of procedural knowledge.

We evaluate TokMem in two complementary settings. First, in atomic recall, we treat each of the 1,000 tasks from Super-Natural Instructions (Wang et al., 2022a) as a distinct procedure. In this setting, TokMem successfully stores and retrieves these procedures efficiently without catastrophic forgetting, outperforming retrieval-based baselines by eliminating the noise and computational overhead of text-based prompts. Second, in compositional recall using function-calling benchmark (Liu et al., 2024b), each tool invocation is treated as an atomic procedure, and answering a query requires chaining multiple such procedures together. Here, TokMem consistently matches or surpasses the performance of parametric fine-tuning while using significantly fewer trainable parameters.

## 2  METHOD

In this section, we first review the Transformer processing pipeline and the limitations of current textualized prompting (§ 2.1). We then introduce TokMem, detailing how procedures are encoded into a single memory token (§ 2.2), how these tokens are recalled and chained at inference time (§ 2.3), and the renormalization strategy used to stabilize the memory bank (§ 2.4).

## 2.1 TEXTUALIZED CONTEXT ENGINEERING

A Transformer (Vaswani et al., 2017) processes a sequence of tokens $(a_1, \ldots, a_n) \in \mathbb{N}^n$, where each $a_i$ is an integer representing the index of a token (usually a sub-word). The model retrieves the corresponding embedding vector from the embedding layer and produces an input sequence $(\boldsymbol{x}_1, \ldots, \boldsymbol{x}_n)^\top \in \mathbb{R}^{n \times d}$, which is then used to predict the next token in sequence.

Recent advances in prompting can be viewed as *textualized context engineering*, where the goal is to choose input tokens that steer the model toward desired behavior. For example, chain-of-thought prompting (Wei et al., 2022a) augments the input with intermediate reasoning steps to improve logical inference. Similarly, retrieval-based methods like RAG (Lewis et al., 2020) fetch external documents to ground generation, while recent memory systems such as MemGPT (Packer et al., 2024) retrieve text history into the active context to maintain conversational state. Critically, these approaches all rely on expanding the *textual* prompt, consuming the limited context window and increasing inference latency due to the quadratic cost of self-attention.

## 2.2 TOKMEM: PROCEDURAL MEMORY AS A TOKEN

Our key idea is that frequently reused task procedures can be effectively "compressed" and stored by encoding them into a dedicated memory token, significantly reducing the verbosity of task instructions. Consider $l$ memory tokens. They are added to the vocabulary as special tokens, each represented by an embedding. Thus, they form a memory bank of $l$ special embeddings:

$$M = \begin{bmatrix} \boldsymbol{m}_1^\top \\ \vdots \\ \boldsymbol{m}_l^\top \end{bmatrix} \in \mathbb{R}^{l \times d}, \qquad \boldsymbol{m}_i \in \mathbb{R}^d. \tag{1}$$

Each $\boldsymbol{m}_i$ is a trainable vector with no direct textual form and represents a unique procedure. For simplicity, we assign each memory token $\boldsymbol{m}_i$ a special token index $a_{m_i} \in \mathbb{N}$.

To connect these tokens with training, we first describe a single training instance. We define a *procedure–response* pair, where a procedure is invoked by a memory token $\boldsymbol{m}_i$ and the response is a sequence of textual tokens $(r_{i1}, \ldots, r_{in}) \in \mathbb{N}^n$ that realizes the procedure in text. For example, the response may specify tool-call arguments or follow a particular output format implied by the procedure.

We formulate training as a supervised task where each procedure–response pair is provided, represented by concatenating a procedural memory token $a_{m_i}$ with the embeddings of $(r_{i1}, \ldots, r_{in})$. Each training instance may contain multiple procedure–response turns, modeling tasks that require multi-step reasoning or composition for a query $q$. Formally, the sequentialized training sequence has the layout

$$\boldsymbol{a} = \big(q_1, \ldots, q_k, \underbrace{a_{m_i}, a_{r_{i1}}, a_{r_{i2}}, \ldots}_{\text{procedure–response pair}}, \underbrace{a_{m_j}, a_{r_{j1}}, a_{r_{j2}}, \ldots}_{\text{procedure–response pair}}, \ldots \big). \tag{2}$$

We adopt the standard next-token prediction loss:

$$\mathcal{L}(\boldsymbol{a}; M) = -\sum_{i > k} \log \Pr(a_i \mid \boldsymbol{a}_{<i}; M). \tag{3}$$

The memory embeddings $(\boldsymbol{m}_1, \ldots, \boldsymbol{m}_l)$ are shared between the input embedding layer and the LM head. During optimization, these embeddings are trainable, whereas the pre-trained text token embeddings and the backbone LLM remain frozen. Across the training corpus, each memory embedding is exposed to diverse queries and responses, allowing it to learn the underlying procedure representation. We visualize the training process in Figure 1a.

## 2.3 INFERENCE WITH MEMORY TOKENS

At inference time, TokMem recalls procedures via memory routing and conditional generation, where *routing* selects the appropriate memory token for a query. Given $q = (q_1, \ldots, q_k)$, the model predicts a distribution over memory tokens from the final hidden state $h_k$:

$$P(a_{m_i} \mid q) \propto \exp \big( \text{logit}(m_i \mid h_k) \big), \tag{4}$$

We select the most probable token $a_{m^*}$ and append it to form $[q \, ; \, a_{m^*}]$, after which the model generates the response auto-regressively. For multi-procedure queries, the model can predict additional memory tokens after generating each response segment, enabling token chaining as in Figure 1b. If a query does not match any learned procedure, all memory-token logits may remain low and the model defaults to generating regular text.

In summary, the model decides whether (and how many) memory tokens to recall based on the training layout in (Eq. 2).

## 2.4 STABILIZING NEW MEMORIES

While TokMem supports the modular addition of procedures, integrating new tokens into a frozen backbone presents stability challenges. In continual learning settings, newly optimized memory embeddings are prone to norm inflation (Hou et al., 2019). These new embeddings can dominate the routing logits, suppressing the retrieval of old memories regardless of semantic relevance.

To address this, we introduce renormalization, a lightweight post-update calibration of the memory bank $M \in \mathbb{R}^{l \times d}$. Let $A$ and $I$ denote the indices of the new and existing procedural memories, respectively. We estimate the typical scale from the existing set:

$$\bar{n}_I \;=\; \mathrm{mean}_{j \in I} \big\| \boldsymbol{m}_j \big\|_2, \tag{5}$$

and rescale each new embedding as

$$\boldsymbol{m}_i \;\leftarrow\; \boldsymbol{m}_i \cdot \frac{\bar{n}_I}{\|\boldsymbol{m}_i\|_2 + \varepsilon}, \quad i \in A. \tag{6}$$

This operation preserves the directions of newly added embeddings while aligning their magnitudes to the established scale of the memory bank, ensuring smooth integration without overwhelming routing dynamics. The computational overhead is negligible, scaling as $O(|A|d)$.

## 3 EXPERIMENTS

We evaluate TokMem in two complementary scenarios. In *atomic memory recall*, each task from Super-Natural Instructions (Wang et al., 2022a) is treated as a standalone procedure, where a query maps directly to the desired response. In *compositional memory recall*, we evaluate multi-procedure tool use on a function-calling dataset (Liu et al., 2024b): each tool invocation is modeled as an atomic procedure, and solving a query requires composing multiple calls. Experiments are conducted on the Qwen (Qwen Team, 2025) and Llama (Llama Team, 2024) families, ranging from 0.5B to 8B.

## 3.1 EXPERIMENTAL SETUP

**Baselines.** Across both settings, we compare TokMem against textualized context engineering, retrieval-augmented memory, and parameter-efficient fine-tuning baselines.

- **Base**: In the atomic setting, the model answers queries without demonstrations, providing an empirical lower bound that highlights the need to recall task knowledge.

- **ICL**: In the compositional setting, we augment the input with all tool descriptions and prepend two compositional procedure–response demonstrations as a context-engineering baseline.

- **RAG**: We use Sentence-BERT (Reimers & Gurevych, 2019) to retrieve relevant demonstrations or tool-use examples and prepend them to the query, following memory-augmented generation (Packer et al., 2024; Chhikara et al., 2025; Xu et al., 2025).

- **Fine-tuning**: We fine-tune low-rank adapters (LoRA; Hu et al., 2022), which are inserted into the query and value projections of the transformer, with a parameter count comparable to TokMem. This provides a parametric form of procedural memory.

- **Replay Memory**: To mitigate catastrophic forgetting during fine-tuning, we adopt experience replay (Mnih et al., 2015) by maintaining a buffer of previously seen tasks or tools and mixing them into the current training data.

Table 1: Atomic recall performance on SNI (ROUGE-L). TokMem consistently outperforms fine-tuning and RAG across models and scales, maintaining strong performance even at 1,000 tasks.

| Model | Method | Number of Tasks | | | | | Avg. |
|---|---|---|---|---|---|---|---|
| | | 10 | 50 | 200 | 500 | 1000 | |
| Qwen 2.5 0.5B | *Base* | 33.9 | 39.0 | 38.8 | 39.1 | 38.5 | 37.9 |
| | *RAG* | 50.4 | 43.2 | 38.8 | 36.2 | 34.7 | 40.7 |
| | *Fine-Tuing* | 52.4 | 48.0 | 40.6 | 41.7 | 43.2 | 45.2 |
| | *Replay Memory* | 52.4 | 49.5 | 47.2 | 47.7 | 46.7 | 48.7 |
| | *TokMem* | 52.8 | **51.3** | 49.3 | 50.2 | **50.0** | 50.7 |
| | *TokMem+DC* | **53.8** | 50.5 | **50.2** | **50.9** | **50.0** | **51.1** |
| Llama 3.2 3B | *Base* | 16.6 | 19.9 | 20.0 | 18.7 | 18.2 | 18.7 |
| | *RAG* | 60.0 | 48.7 | 45.8 | 42.3 | 39.9 | 47.3 |
| | *Fine-Tuing* | 67.1 | 59.1 | 59.5 | 58.4 | 57.9 | 60.4 |
| | *Replay Memory* | 67.1 | 61.1 | 60.6 | 61.4 | 60.0 | 62.0 |
| | *TokMem* | 68.0 | 62.3 | **61.2** | 61.5 | **61.5** | **62.9** |
| | *TokMem+DC* | **68.8** | **62.5** | 58.7 | **61.7** | 61.1 | 62.6 |
| Llama 3.1 8B | *Base* | 27.2 | 27.8 | 30.4 | 29.6 | 29.5 | 28.9 |
| | *RAG* | 63.8 | 53.9 | 49.1 | 45.3 | 42.6 | 50.9 |
| | *Fine-Tuing* | 75.8 | 64.3 | 63.2 | 58.7 | 61.6 | 64.7 |
| | *Replay Memory* | 75.8 | 65.2 | 64.5 | 63.4 | 63.6 | 66.5 |
| | *TokMem* | 75.4 | 65.5 | **65.1** | **64.4** | **64.8** | **67.0** |
| | *TokMem+DC* | **75.6** | **65.8** | 63.7 | 64.2 | 64.4 | 66.7 |

**Training Details.** All methods are implemented in HuggingFace Transformers and trained on a single NVIDIA A6000 GPU (48GB) using mixed-precision (bfloat16) training. The backbone models remain frozen: for fine-tuning, only the adapter weights (rank $r = 8$) are updated, whereas for TokMem, only the embeddings of newly added procedure IDs are trainable. We expand the tokenizer vocabulary with these procedure IDs and initialize their embeddings by averaging the pretrained embeddings (Hewitt, 2021). For Replay Memory, we mix 20% replayed samples into each batch, using a buffer of 500 examples refreshed every 10 tasks in the atomic setting and 1,000 examples updated each round in the compositional setting.

We optimize with AdamW using a learning rate of $5 \times 10^{-5}$ for fine-tuning and $5 \times 10^{-3}$ for TokMem; weight decay is $10^{-2}$ for fine-tuning and zero for TokMem. Training runs for one epoch with batch size 4 and maximum sequence length 1024, using teacher forcing and applying the loss only to memory-token and response positions.

**Evaluation Metrics.** We evaluate TokMem from two perspectives: (1) memory-token routing accuracy, which measures whether the correct memory tokens are selected, and (2) task performance, which measures task-level generation quality (e.g., ROUGE-L and F1).

## 3.2 ATOMIC MEMORY RECALL

**Dataset Details.** We evaluate on Super-Natural Instructions (SNI; Wang et al., 2022a), which provides diverse QA-style natural language tasks. We use ROUGE-L (Lin, 2004) to measure generation quality. Each task is treated as an individual procedure: a query directly invokes the learned procedure to produce the desired response. We sample 1,000 English tasks; each task contains 500 training samples and 50 test samples. To reflect how memories are typically acquired over time, we introduce tasks sequentially during training rather than all at once, while shuffling examples within each task. We scale the number of tasks from 10 to 1,000 and record checkpoints after training on $\{10, 50, 200, 500, 1,000\}$ tasks. This resembles incremental domain adaptation (Asghar et al., 2020), where at each checkpoint we evaluate performance across all previously seen tasks. Additional task details are provided in Appendix A.1.

**Results and Findings.** As shown in Table 1, baseline methods such as Base are stable but fail to achieve competitive performance. RAG performs reasonably well when the memory load is small, but degrades quickly as the number of stored task memories increases, indicating its sensitivity to

Table 2: Task routing accuracy. TokMem achieves near-perfect routing accuracy at 1,000 tasks, far exceeding the RAG retriever, whose accuracy falls below 80%.

| Model | Method | Number of Tasks | | | | |
|---|---|---|---|---|---|---|
| | | 10 | 50 | 200 | 500 | 1000 |
| Sentence-BERT | *RAG* | 99.6 | 92.6 | 88.7 | 83.2 | 79.7 |
| Qwen 2.5 0.5B | *TokMem* | 99.4 | 98.6 | 97.4 | 96.9 | 94.7 |
| | *TokMem+DC* | **99.4** | **99.2** | **98.4** | **97.2** | **96.1** |
| Llama 3.2 3B | *TokMem* | **100.0** | **99.9** | **98.3** | **97.1** | **96.1** |
| | *TokMem+DC* | 99.8 | 99.3 | 97.2 | 96.2 | 95.4 |
| Llama 3.1 8B | *TokMem* | **99.8** | **99.6** | **98.9** | **97.7** | **97.5** |
| | *TokMem+DC* | 99.7 | 99.4 | 97.8 | 97.2 | 97.2 |

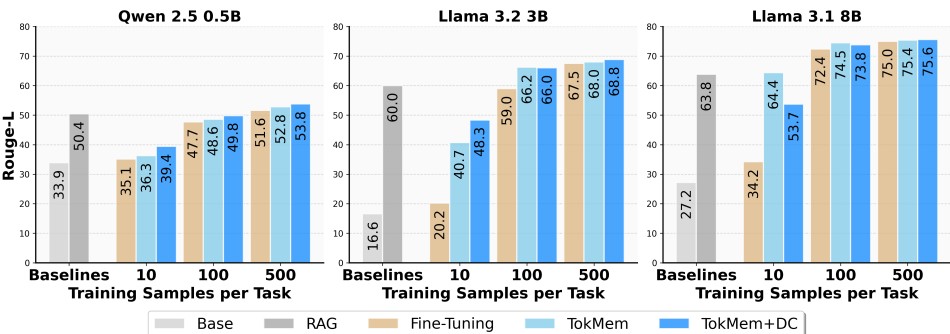

Figure 2: Sample efficiency on a 10-task mixture from SNI. TokMem consistently outperforms fine-tuning in the low-data regime and can surpass RAG with only 10 training samples, demonstrating strong few-shot capability.

the retriever's quality. Parametric methods such as fine-tuning achieve stronger initial accuracy but suffer from forgetting as tasks accumulate; replay memory alleviates this issue but still falls short of TokMem. By contrast, TokMem maintains high accuracy with minimal performance drop as it acquires new task memories, achieving the best average results across all settings.

We also include a variant where each memory token is decoupled into an address token and a steering token. This decoupling separates the roles of memory tokens and increases the capacity of TokMem. We refer to this variant as TokMem with decoupled embeddings (TokMem+DC); further details are provided in Appendix A.2.

The decoupled variant (TokMem+DC) yields modest gains for the smaller Qwen-0.5B model but no improvement for larger Llama models, and in some cases it underperforms TokMem when scaling to many tasks. Overall, although TokMem+DC is a tempting variant, it does not provide additional benefits. We therefore focus on the simple yet effective TokMem (without DC).

Table 2 further highlights TokMem's robustness in memory routing. Its accuracy remains above 94% even at 1,000 tasks with the smallest 0.5B model, significantly outperforming the Sentence-BERT retriever used in RAG. By contrast, the retriever's accuracy drops below 80% when routing over 1,000 tasks. This high-fidelity routing enables TokMem to sustain strong performance without external retrieval or fine-tuning, demonstrating its advantage for continual and large-scale task acquisition.

**Analysis of Training Sample Efficiency.** We compare the training sample efficiency of LoRA fine-tuning and TokMem on 10 SNI tasks. We vary the number of training samples per task from 10 (few-shot) to 500, and report performance across these sample budgets in Figure 2. We set the adapter rank to $r = 1$, which helps reduce overfitting in the low-data regime and matches the parameter scale of TokMem. Results show that TokMem consistently outperforms fine-tuning across all budgets, with the largest gains at small sample sizes. The decoupled variant (TokMem+DC) yields modest gains in this controlled setting, mainly at higher sample budgets. Overall, these results

Table 3: Compositional tool-use performance on APIGen. TokMem achieves strong tool selection and argument F1 across multiple tool calls, outperforming ICL and RAG while requiring less input augmentation, and surpassing fine-tuning with far fewer trainable parameters.

| Model | Method | #Params | Tool Selection | | | | Argument | | | |
|---|---|---|---|---|---|---|---|---|---|---|
| | | | 2 calls | 3 calls | 4 calls | Avg. | 2 calls | 3 calls | 4 calls | Avg. |
| Llama 3.2 1B | ICL | – | 27.6 | 11.1 | 10.5 | 16.4 | 0.6 | 0.7 | 0.0 | 0.4 |
| | RAG | – | 29.5 | 10.8 | 10.5 | 16.9 | 7.2 | 1.0 | 0.0 | 2.7 |
| | Fine-Tuing | 0.85M | 10.4 | 9.5 | 7.0 | 9.0 | 77.3 | 72.6 | 55.8 | 68.6 |
| | TokMem (w/o adapt) | 0.10M | 86.8 | 80.9 | 90.8 | 86.2 | 68.9 | 61.1 | 73.0 | 67.7 |
| | TokMem (w/ adapt) | 0.10M | 98.4 | 98.0 | 98.9 | **98.4** | 84.3 | 84.3 | 87.8 | **85.5** |
| Llama 3.2 3B | ICL | – | 66.8 | 59.2 | 59.6 | 61.9 | 42.2 | 42.3 | 38.8 | 44.1 |
| | RAG | – | 78.1 | 71.2 | 69.3 | 72.8 | 54.8 | 53.1 | 62.7 | 56.9 |
| | Fine-Tuing | 2.29M | 98.7 | 98.1 | 96.8 | 97.9 | 87.9 | 86.6 | 82.9 | 85.8 |
| | TokMem (w/o adapt) | 0.15M | 82.6 | 79.3 | 67.2 | 76.4 | 65.4 | 57.2 | 50.2 | 57.6 |
| | TokMem (w/ adapt) | 0.15M | 99.2 | 98.2 | 100.0 | **99.2** | 85.9 | 86.7 | 88.3 | **86.3** |
| Llama 3.1 8B | ICL | – | 79.7 | 72.9 | 75.4 | 76.0 | 51.5 | 52.6 | 57.3 | 53.8 |
| | RAG | – | 79.6 | 75.3 | 93.0 | 82.6 | 53.3 | 57.1 | 69.2 | 59.9 |
| | Fine-Tuing | 3.41M | 98.8 | 97.2 | 98.2 | 98.1 | 87.7 | 86.8 | 88.2 | 87.6 |
| | TokMem (w/o adapt) | 0.20M | 84.9 | 82.0 | 81.6 | 82.8 | 65.8 | 56.7 | 65.9 | 62.8 |
| | TokMem (w/ adapt) | 0.20M | 99.4 | 97.9 | 100.0 | **99.1** | 88.1 | 86.5 | 93.4 | **89.3** |

highlight TokMem's ability to acquire new procedures with limited data, making it both parameter- and data-efficient.

## 3.3 COMPOSITIONAL MEMORY RECALL

**Dataset Details.** We construct a benchmark from the APIGen dataset (Liu et al., 2024b) by sampling 50 frequently used tools. Each tool invocation is treated as an atomic procedure, and solving a query requires composing multiple such procedures. We synthesize 5,000 training queries and 500 test queries, each capped at four tool calls. Additional dataset details are provided in Appendix A.3.

We report performance using two F1 metrics: (1) Tool Prediction F1, which measures whether the model invokes the correct tools; and (2) Argument Generation F1, which evaluates the correctness of tool-call arguments. To account for semantic equivalence, both gold and predicted outputs are normalized into Abstract Syntax Trees before scoring (Patil et al., 2025).

**Adaptation for Compositionality.** We find that TokMem benefits from a brief adaptation phase that exposes the backbone to the compositional structure of memory tokens. Concretely, we fine-tune the backbone on a held-out auxiliary tool set using the same LoRA setup as the baseline. We then merge the adapted weights, after which the backbone remains frozen for memory acquisition and evaluation (see Appendix A.4 for details). Note that the auxiliary tool set is different from the evaluation set, so this procedure does not violate our frozen-backbone setup.

This adaptation teaches the model how to interleave responses with memory tokens. We therefore treat it as part of TokMem for compositional tasks, and use TokMem with adaptation as the default configuration.

**Results and Findings.** Table 3 shows that TokMem achieves the strongest overall performance. Even without adaptation for compositionality, it outperforms RAG while avoiding the added complexity of an external retrieval mechanism. ICL and RAG perform poorly on both tool prediction and argument generation, especially with the smaller Llama 1B model, likely due to its weaker instruction-following ability.

TokMem consistently matches or exceeds LoRA fine-tuning's performance while requiring an order of magnitude fewer trainable parameters. For example, when applied to Llama 8B, LoRA requires 3.41M trainable parameters, whereas TokMem needs only 0.2M and achieves higher performance.

Notably, TokMem exhibits stronger correlation between tool selection and argument generation: improvements in the former translate directly into gains in the latter. By contrast, LoRA fine-tuning shows weaker alignment. For example, on the 1B model, it often generates plausible arguments

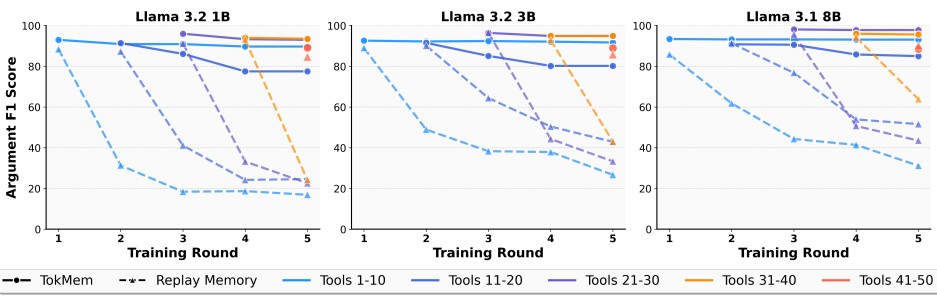

Figure 3: Forgetting analysis under continual adaptation. As new tools are introduced, fine-tuning with replay memory suffers sharp performance drops on earlier tasks, whereas TokMem remains stable. Larger models exhibit stronger retention, likely due to their greater capacity.

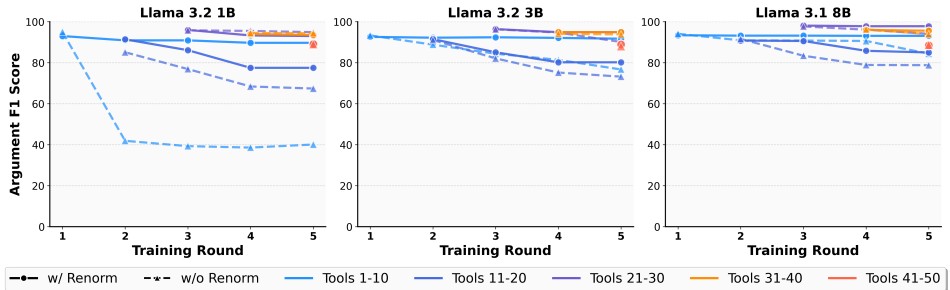

Figure 4: Effect of renormalization on TokMem. Without renormalization, newly learned tokens dominate and older tokens are forgotten, particularly in smaller models with limited embedding capacity. Renormalization mitigates this by balancing token embedding norms.

even when tool selection is incorrect, suggesting that argument generation is not properly grounded in the chosen tool.

TokMem also exhibits stronger compositional generalization than fine-tuning, successfully executing queries with more function calls than it has seen during training. Detailed results are provided in Appendix B.1.

**Analysis of Forgetting.** We compare TokMem with replay memory in a continual adaptation setting, where tools are introduced sequentially over five training rounds (e.g., tools 1–10 in the first round, 11–20 in the second, and so forth). As shown in Figure 3, replay memory struggles to prevent catastrophic forgetting as new tools are introduced. By contrast, TokMem maintains higher performance across tool groups, with only mild declines that mainly reflect the increasing number of tools. Larger models exhibit better retention for both approaches, likely due to their greater capacity, which reduces interference with previously learned tools.

We further investigate the effect of the renormalization step introduced in Section 2.4. Without renormalization, the norms of newly added memory tokens can dominate older tokens in the softmax (see Appendix B.2). As shown in Figure 4, TokMem without renormalization exhibits noticeable forgetting, especially for smaller models. Larger models are more robust even without renormalization, likely due to their greater embedding capacity. Overall, renormalization improves TokMem's resistance to forgetting by balancing routing between new and old memory tokens. We provide further analysis in Appendix B.3 on how keeping the backbone frozen helps prevent forgetting during continual memory acquisition.

## 3.4 ANALYSIS ON MEMORY PLACEMENT

An important design choice in TokMem is where memory tokens are placed within the input sequence, as this directly affects how the backbone model attends to and integrates procedural knowl-

Table 4: Comparison of TokMem and prefix tuning for memorizing text from the *Fanfics* dataset, evaluated by perplexity. The suffix "-k" refers to the number of embeddings, and Steps@90%Best represents the training steps needed to reach 90% of the lowest achieved perplexity. TokMem converges faster and achieves lower perplexity than prefix tuning, especially with a small number of memory tokens.

| Method | 1024 tokens | | 2048 tokens | | 4096 tokens | |
|---|---|---|---|---|---|---|
| | Steps@90%Best ↓ | PPL ↓ | Steps@90%Best ↓ | PPL ↓ | Steps@90%Best ↓ | PPL ↓ |
| *Prefix tuning-1* | 1700 | 3.81 | 2300 | 8.77 | 2200 | 14.32 |
| *TokMem-1* | **1200** | **3.28** | **1400** | **7.07** | **1700** | **12.27** |
| *Prefix tuning-2* | **500** | 1.13 | 1700 | 3.51 | 1800 | 8.38 |
| *TokMem-2* | 600 | **1.09** | **1300** | **2.75** | **1700** | **7.21** |
| *Prefix tuning-5* | 300 | 1.07 | **500** | 1.17 | **1400** | 2.39 |
| *TokMem-5* | **200** | **1.03** | **500** | **1.15** | **1400** | **1.91** |

edge. Although TokMem introduces a memory-routing mechanism for generating tokens, its effectiveness also depends on the placement strategy. Without routing, TokMem reduces to a prompt-tuning method (Li & Liang, 2021; Lester et al., 2021) with learnable embeddings; however, it adopts an *infix* placement: `query ⊕ MEM ⊕ response`. It remains unclear whether this placement is consistently better than the more common *prefix* formulation, `MEM ⊕ query ⊕ response`, used in prior prompt-tuning work, where prefix tokens influence generation before observing the query. To study the impact of placement, we compare prefix and infix placements under matched token budgets in the single-task setting.

**Setups.** We compare TokMem with infix memory placement against prefix tuning by stress-testing the capacity of memory tokens (Sastre & Rosá, 2025) on the recent *Fanfics* dataset, which was collected after the pretraining of the LLMs (Kuratov et al., 2025). We fix the sequence length to 128 tokens and vary the batch size from 8 to 32, compressing batches of 1024 to 4096 response tokens into 1 to 5 memory tokens. For each sequence, we prepend a randomly generated query that serves only as a marker to distinguish the two placements; the target to be learned remains the response.

We measure learning speed using Steps@90%Best, defined as the number of training steps (evaluated every 100 steps) required to reach 90% of the best perplexity. Results are averaged over five runs. Additional experiments on generalization to a math reasoning dataset are provided in Appendix B.4.

**Results.** Table 4 shows that TokMem consistently achieves lower perplexity and often converges faster than prefix tuning. With a single token, TokMem reaches 90% of its best perplexity roughly 30% sooner than prefix tuning, suggesting that conditioning on memory after the query helps the model learn more efficiently. Interestingly, when more tokens are available (e.g., five tokens), the performance gap narrows. This suggests that prior work (Li & Liang, 2021), which typically uses dozens or even hundreds of tokens, may have underestimated the importance of memory placement in low-token regimes, where each token must compress more procedural information.

## 4 RELATED WORK

Equipping LLMs with memory has been explored along multiple directions. Most existing approaches emphasize declarative memory, where the objective is to store and retrieve explicit information such as facts and conversation history (Packer et al., 2024; Chhikara et al., 2025; Wang et al., 2024). By contrast, parameter-based approaches internalize task-specific behaviors within model parameters, resembling procedural memory. TokMem builds on this latter view while emphasizing modularity and compositionality.

**Text-Based External Memory.** A common approach is to externalize memory as textual content retrieved at inference time. Retrieval-augmented generation (RAG; Lewis et al., 2020) and its variants (Guu et al., 2020; Karpukhin et al., 2020; Borgeaud et al., 2022; Khandelwal et al., 2020) attach relevant textual chunks during inference, while RET-LLM (Modarressi et al., 2024) encodes

knowledge as symbolic triplets. Building on these ideas, more recent systems extend these ideas to conversational settings through hierarchical or summarization-based memory states; examples include MemGPT (Packer et al., 2024), Mem0 (Chhikara et al., 2025), and A-Mem (Xu et al., 2025). While effective for factual recall, these approaches are not optimized for procedural control and often incur significant inference-time overhead due to repeatedly re-reading textual memory.

**Parameter-Based Memory.** Another line of work encodes memory directly into model parameters. Fine-tuning and multitask instruction tuning (Wei et al., 2022a; Sanh et al., 2022) allow models to acquire new procedures, but task knowledge often becomes entangled within the shared weights. Parameter-efficient fine-tuning methods such as LoRA (Hu et al., 2022) and Flora (Hao et al., 2024) disentangle new knowledge from the pre-trained base model, but still suffer from interference when learning multiple tasks. Mixture-of-LoRAs (Feng et al., 2024) address this entanglement issue by dynamically routing inputs to task-specific experts; but the mixtures are typically invoked independently and are not designed for memory composition. MemoryLLM (Wang et al., 2024) introduces latent memory pools but remains entangled. Prompt-based methods such as prompt tuning (Lester et al., 2021; Wu et al., 2024; 2026) store knowledge implicitly as global embeddings without selective routing, and L2P (Wang et al., 2022b) introduces modular prompt pools but still relies on an external controller to determine which prompts are retrieved. Prompt compression methods (Mu et al., 2023; Chevalier et al., 2023) compress prompts into context-agnostic representations, which may distort prompt information. ToolGen (Wang et al., 2025) compresses tools into virtual tokens but focuses on post-training the backbone through multi-stage fine-tuning. By contrast, our Tok-Mem keeps the backbone frozen and introduces dedicated memory embeddings that can be added or composed without retraining, supporting continual adaptation.

**Compositional Memory.** A complementary direction explores how models compose skills from simpler building blocks. Chain-of-thought prompting (Wei et al., 2022b) and tool-augmented reasoning frameworks such as Toolformer (Schick et al., 2023) enable multi-step reasoning, but they rely on textual instructions that must be re-interpreted at each step. Modular parameter methods (Rosenbaum et al., 2018; Pfeiffer et al., 2021) create specialized adapters that can be recombined, but composition typically requires parameter merging or heuristic routing. TokMem differs by representing procedures as discrete tokens that can be chained directly in generation, enabling lightweight, parameter-isolated composition.

## 5 Conclusion

We introduced TokMem, a parameter-efficient framework that encodes procedural memory as compact tokens. TokMem enables selective recall and compositional use of procedures without modifying backbone parameters, achieving strong performance across multitask and tool-augmented reasoning benchmarks.

Future directions are discussed in Appendix C, including the automation of query–procedural decomposition for synthesizing training trajectories, reinforcement learning for stronger compositional generalization and personalization through user-specific memory banks. These extensions pave the way for scalable, compact, and user-adaptive memory systems in large language models.

## Acknowledgments

We thank the reviewers and chairs for their efforts. The research is supported in part by the Natural Sciences and Engineering Research Council of Canada (NSERC), the Amii Fellow Program, the Canada CIFAR AI Chair Program, a donation from DeepMind, and the Digital Research Alliance of Canada (alliancecan.ca).

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

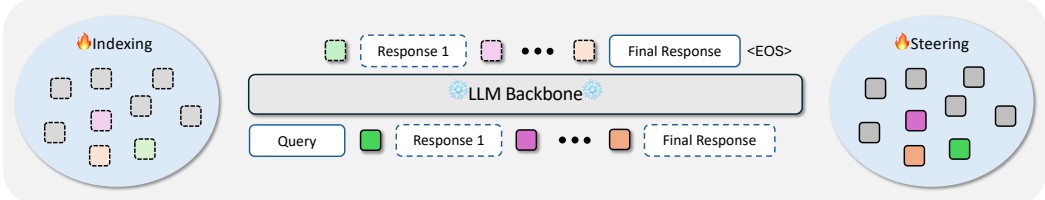

Figure 5: Overview of the decoupled variant of TokMem embeddings, which learn separate memory matrices for memory indexing and generation steering.

# A    EXPERIMENTAL DETAILS

## A.1    DETAILS OF SUPER-NATURAL INSTRUCTIONS

We sample $1{,}000$ English tasks from the SNI dataset, where each task is labeled with a task ID and a short descriptive name. The full list of sampled tasks is provided in Figure 8. Tasks are introduced to the model sequentially in ascending order of their IDs, for instance, the model first sees task 1, then task 2, and so on.

After training on the first $k$ tasks, we save a checkpoint and evaluate performance on the test set of each of the $k$ tasks encountered so far. This simulates a continual learning setup where the model is expected to acquire new procedures while retaining previously learned ones. Once the model has been trained on all $1{,}000$ tasks, it is expected be able to perform all of them without forgetting earlier tasks.

## A.2    DETAILS OF DECOUPLED EMBEDDING FOR TOKMEM

In the standard TokMem formulation, each memory token embedding $\boldsymbol{m}_i \in \mathbb{R}^d$ is shared across two roles: (1) indexing for memory routing and (2) condition to steer for generation. We consider a decoupled (DC) variant that separates these functions into two embedding matrices:

$$M^{\text{index}} = (\boldsymbol{u}_1; \ldots; \boldsymbol{u}_l) \in \mathbb{R}^{l \times d}, \qquad M^{\text{steer}} = (\boldsymbol{s}_1; \ldots; \boldsymbol{s}_l) \in \mathbb{R}^{l \times d}. \tag{7}$$

As seen in Figure 5, $M^{\text{index}}$ provides *index embeddings* at the output layer. The model produces a distribution over indices $i$ according to $M^{\text{index}}$ when a memory token is predicted. The chosen index $i$ is then used to retrieve the corresponding steering embedding $\boldsymbol{s}_i$ from $M^{\text{steer}}$, which is then appended into the sequence and conditions subsequent generation.

Training follows the standard next-token prediction objective, analogous to Equation 3:

$$\mathcal{L}(\boldsymbol{a}; M^{\text{index}}, M^{\text{steer}}) = -\sum_{i > k} \log \Pr(a_i \mid \boldsymbol{a}_{<i}; M^{\text{index}}, M^{\text{steer}}). \tag{8}$$

where $k$ denotes the query length. During optimization, only $M^{\text{index}}$ and $M^{\text{steer}}$ are updated, while the backbone remains frozen. In addition, the renormalization treatment introduced in Section 2.2 is applied only to the index embeddings $M^{\text{index}}$.

This decoupled formulation provides a clean separation of functionality: routing is handled via $M^{\text{index}}$, while steering is controlled by $M^{\text{steer}}$. However, our experiments do not show consistent improvements over the coupled formulation, particularly on larger models where their larger embedding capacity is sufficient to jointly support both roles. We therefore advocate for the simple yet effective TokMem (without DC).

## A.3    DETAILS OF FUNCTION-CALLING DATASET

To evaluate compositional memory recall, we sample 50 tools from the APIGen dataset (Liu et al., 2024b). The list of tools and their corresponding descriptions is provided in Table 7.

---

**Algorithm 1** Adaptation Phase for Compositional Memory Recall

---

**Require:** Pre-trained backbone $f_{\theta_0}$, adaptation traces $\mathcal{D}_{\text{adapt}}$ from held-out procedures
 1: Initialize the backbone $\theta \leftarrow \theta_0$ and temporary memory embeddings $\mathcal{M}$
 2: Set learning rates $\eta_\theta$ and $\eta_\mathcal{M}$
 3: **for** each minibatch in $\mathcal{D}_{\text{adapt}}$ **do**
 4:     Insert $\mathcal{M}$ into the sequence; run a forward pass with $f_\theta$
 5:     Compute the loss $\mathcal{L}$ on memory and response tokens
 6:     $\theta \leftarrow \theta - \eta_\theta \nabla_\theta \mathcal{L}$
 7:     $\mathcal{M} \leftarrow \mathcal{M} - \eta_\mathcal{M} \nabla_\mathcal{M} \mathcal{L}$
 8: **end for**
 9: Discard the temporary memory $\mathcal{M}$ and freeze the backbone $\theta$
10: **return** the adapted backbone $f_\theta$

---

For each tool, we collect 50 query–call pairs, some of which may involve multiple calls to the same tool. We categorize the use of the same tool as non-compositional tool use, yielding a total of $50 \times 50 = 2{,}500$ samples.

Additionally, we augment the dataset by synthesizing complex queries by combining calls across different tools. These multi-step queries require the model to invoke multiple tools in sequence. To avoid data leakage, we split these samples into training and test sets, capping the synthesized samples at 5,000 for training and 500 for testing.

### A.4 DETAILS OF ADAPTATION PHASE

In compositional scenarios, the model should not only recall individual procedures but also compose them to solve multi-step queries. To prepare TokMem for such use, we construct a held-out auxiliary training set comprising an additional 50 tools (5,000 samples) that is disjoint from the tool set detailed in Appendix A.3. The backbone is then fine-tuned for one epoch on this set using LoRA, jointly with the temporary memory embeddings.

The intuition behind this adaptation phase is to let the base LLM learn to use the embeddings with compositional memory recall, rather than memorizing on the tool descriptions themselves. After adaptation, the temporary embeddings are discarded, while the adapted backbone is retained for training on new tools. This procedure provides a general inductive bias for modular composition, enabling the model to generalize to new tools and procedures without further retraining.

Algorithm 1 summarizes this lightweight procedure. Temporary memory embeddings are inserted into the input sequence, the loss is optimized jointly over memory and response tokens, and the temporary memory bank is discarded once the backbone has adapted.

## B ADDITIONAL RESULTS

### B.1 RESULTS ON COMPOSITIONAL GENERALIZATION

We observe that TokMem provides clear advantages in compositional generalization over fine-tuning. Table 5 reports Argument F1 when the Llama 3B model is evaluated on queries that require more function calls than those observed during training.

Notably, when trained solely on single-call data, TokMem achieves much stronger performance than fine-tuning when evaluated on test data with 2–4 calls. This demonstrates that memory tokens trained for atomic procedures can be effectively composed at test time, enabling strong zero-shot generalization to multi-step behavior.

As the training regime is expanded to include more calls (e.g., up to 3 or 4), the performance gap narrows, but TokMem remains competitive or slightly ahead across configurations. These results suggest that TokMem naturally supports compositionality, enabling flexible chaining of learned procedures without requiring task-specific fine-tuning.

Table 5: TokMem generalizes to longer tool-call chains at test time, significantly outperforming fine-tuning in zero-shot multi-step settings (trained with 1 call and tested with 2–4 calls).

| Train Maximum Calls | Method | Test Time | | | |
| --- | --- | --- | --- | --- | --- |
| | | 2 calls | 3 calls | 4 calls | Avg. |
| 1-call | *Fine-tuning* | 34.9 | 21.3 | 14.1 | 23.4 |
| | *TokMem* | 60.3 | 54.3 | 48.9 | **54.5** (+31.1) |
| 2-call | *Fine-tuning* | 86.2 | 78.8 | 64.8 | 76.6 |
| | *TokMem* | 82.0 | 81.8 | 82.3 | **82.0** (+5.4) |
| 3-call | *Fine-tuning* | 86.9 | 85.5 | 80.3 | 84.2 |
| | *TokMem* | 86.8 | 84.0 | 84.7 | **85.2** (+1.0) |
| 4-call | *Fine-tuning* | 87.9 | 86.6 | 82.9 | 85.8 |
| | *TokMem* | 85.9 | 86.7 | 88.3 | **86.3** (+0.5) |

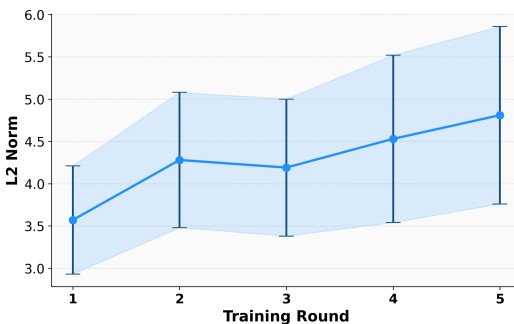

Figure 6: $\ell_2$ norm of newly added memory tokens. Each round introduces 10 new tool memories. Error bars indicate the standard deviation across the 10 tokens added in each round.

## B.2 ANALYSIS OF NORM INFLATION FOR NEWER MEMORIES

We analyze the $\ell_2$ norm of the learned memory embeddings when tools are introduced sequentially with Llama 3.2 3B. We follow the setup in Figure 4 by training TokMem for 5 rounds and adding 10 tools per round without our renormalization treatment detailed in Section 2.4. Figure 6 shows that newly added memory tokens gradually increase their $\ell_2$ norms, which leads to competition with existing frozen tokens in the softmax used for memory routing.

## B.3 ANALYSIS OF UNFREEZING LLM BACKBONE FOR TOKMEM

We further examine the importance of freezing the backbone when adding new tool memories, reflecting real-world usage where procedural knowledge grows incrementally over time. This setting contrasts with recent approaches (Wang et al., 2025) that aim to compress tool use description by post-training the backbone. Such methods rely on modifying backbone parameters, which hinders continual adaptation and risks overwriting prior knowledge.

As shown in Figure 7, We confirm that unfreezing the backbone during TokMem adaptation leads to severe forgetting of previously learned tools. By contrast, our choice of freezing the backbone meaningfully preserves prior capabilities while allowing new tool memories to be incorporated without loss, highlighting TokMem's advantage for incremental adaptation. This analysis suggests that TokMem strikes an effective balance between performance and continual adaptation.

## B.4 ADDITIONAL ANALYSIS ON MEMORY PLACEMENT

In Section 3.4, we have stress-tested the memorization with different memory token placement (prefix vs. infix) by varying numbers of memory tokens. We now turn to the GSM8K math reasoning dataset (Cobbe et al., 2021) to evaluate generalization and training sample efficiency.

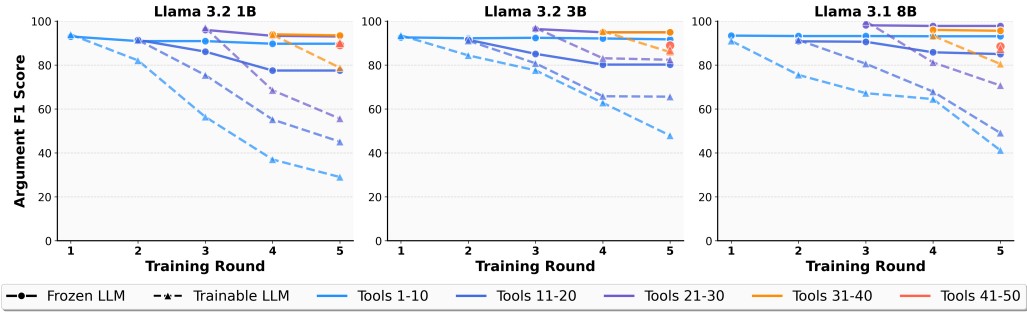

Figure 7: Comparison between freezing and unfreezing the backbone. Allowing the backbone to update when adding new tool memories causes severe forgetting, whereas freezing preserves prior tools while enabling new ones.

Table 6: Comparison of prefix tuning and TokMem conditioning embeddings on GSM8K with two different Llama model sizes. TokMem achieves higher compliance with required output formats and stronger exact-match accuracy than prefix tuning, especially in low-data regimes.

| Data% | Method | Llama 3.2 1B | | Llama 3.2 3B | |
|---|---|---|---|---|---|
| | | Compliance↑ | EM ↑ | Compliance↑ | EM ↑ |
| 20% | *Prefix tuning* | 0.0 | 0.0 | 45.9 | 33.1 |
| | *TokMem* | **98.0** | **37.7** | **94.6** | **65.6** |
| 100% | *Prefix tuning* | 82.8 | 30.0 | 97.2 | 64.1 |
| | *TokMem* | **97.4** | **39.1** | **98.2** | **66.9** |

Our experiments use Llama 3.2 1B and 3B as backbone models and compare prefix tuning against TokMem under two training setups: using only 20% of the training set (a low-data regime) or the full dataset. We report two evaluation metrics.

- **Compliance** measures whether the model follows the required answer format, i.e., producing the final answer after the delimiter "####". This metric isolates procedural-memory recall from the reasoning abilities already present in the backbone.

- **Exact Match (EM)** measures the correctness of the final answer after standard text normalization (e.g., removing commas or extraneous symbols).

As shown in Table 6, TokMem significantly outperforms prefix tuning, particularly in the low-data setting. With only 20% of the data, prefix tuning fails to provide meaningful results, yielding zero compliance and EM on the 1B model and underperforming on the 3B model. By contrast, Tok-Mem achieves near-perfect compliance and substantially higher EM scores across both backbones. When trained on the full dataset, prefix tuning improves considerably, yet TokMem continues to deliver stronger compliance and higher EM, showing its superior data efficiency and more reliable procedural control.

## C FUTURE WORK

Our experiments are conducted on the research-oriented SNI and APIGen datasets, which provide a controlled environment for analyzing atomic and compositional recall. While these settings demonstrate the feasibility and effectiveness of one-token procedural memory without backbone training, they do not fully capture the diversity and ambiguity of real-world procedures.

In particular, richer forms of composition, such as interleaving function calls from APIGen with NLP tasks from SNI, are bottlenecked by TokMem's reliance on manually curated, procedurally decomposed datasets. To bridge this gap, a future direction is the automation of query–procedural decomposition. By leveraging highly capable teacher LLMs to synthesize interleaved training tra-

jectories, we can automatically map complex queries to discrete procedural tokens, providing a scalable alternative to manual annotation.

Beyond automated data synthesis, additional future directions include incorporating reinforcement learning to improve generalization over complex compositional structures, and enabling personalization by allowing users to attach custom, user-specific memory banks while keeping the backbone frozen. These extensions pave the way for scalable, compact, and user-adaptive memory systems in large language models.

## D    USE OF LARGE LANGUAGE MODELS

We used ChatGPT as a general-purpose assistant to improve the writing of our paper, including grammar, readability, and clarity. Additionally, we used it to search for potentially missing related work, which we then manually read and is discussed in the paper. All ideas, analyses, and conclusions presented in this paper are our own, and we take full responsibility for the content.

Table 7: Details (names and descriptions) of the sampled tools from the APIGen dataset.

| ID | Tool | Description |
|---|---|---|
| 1 | auto_complete | Fetch auto-complete suggestions for a given query using the Wayfair API. |
| 2 | binary_addition | Adds two binary numbers and returns the result as a binary string. |
| 3 | binary_search | Performs binary search on a sorted list to find the index of a target value. |
| 4 | cagr | Calculates the Compound Annual Growth Rate (CAGR) of an investment. |
| 5 | calculate_factorial | Calculates the factorial of a non-negative integer. |
| 6 | calculate_grade | Calculates the weighted average grade based on scores and their corresponding weights. |
| 7 | calculate_median | Calculates the median of a list of numbers. |
| 8 | can_attend_all_meetings | Determines if a person can attend all meetings given a list of meeting time intervals. |
| 9 | cosine_similarity | Calculates the cosine similarity between two vectors. |
| 10 | count_bits | Counts the number of set bits (1's) in the binary representation of a number. |
| 11 | create_histogram | Create a histogram based on provided data. |
| 12 | directions_between_2_locations | Fetches the route information between two geographical locations including distance, duration, and steps. |
| 13 | fibonacci | Calculates the nth Fibonacci number. |
| 14 | final_velocity | Calculates the final velocity of an object given its initial velocity, acceleration, and time. |
| 15 | find_equilibrium_index | Finds the equilibrium index of a list, where the sum of elements on the left is equal to the sum of elements on the right. |
| 16 | find_first_non_repeating_char | Finds the first non-repeating character in a string. |
| 17 | find_longest_word | Finds the longest word in a list of words. |
| 18 | find_max_subarray_sum | Finds the maximum sum of a contiguous subarray within a list of integers. |
| 19 | find_minimum_rotated_sorted_array | Finds the minimum element in a rotated sorted array. |
| 20 | flatten_list | Flattens a nested list into a single-level list. |
| 21 | format_date | Converts a date string from one format to another. |
| 22 | generate_password | Generates a random password of specified length and character types. |
| 23 | generate_random_string | Generates a random string of specified length and character types. |
| 24 | get_city_from_zipcode | Retrieves the city name for a given ZIP code using the Ziptastic API. |
| 25 | get_pokemon_move_info | Retrieves information about a Pokémon's move using the PokéAPI. |
| 26 | get_product | Fetches product details from an API using the given product ID. |
| 27 | get_products_in_category | Fetches products in a specified category from the demo project's catalog. |
| 28 | greatest_common_divisor | Computes the greatest common divisor (GCD) of two non-negative integers. |
| 29 | integrate | Calculate the area under a curve for a specified function between two x values. |
| 30 | investment_profit | Calculates the profit from an investment based on the initial amount, annual return rate, and time. |
| 31 | is_anagram_phrase | Checks if two phrases are anagrams of each other, ignoring whitespace and punctuation. |
| 32 | is_leap_year | Checks if a year is a leap year. |
| 33 | is_palindrome | Checks if a string is a palindrome. |
| 34 | is_power | Checks if a number is a power of a given base. |
| 35 | is_rotation | Checks if one string is a rotation of another string. |
| 36 | is_valid_ip_address | Checks if a string is a valid IP address (IPv4). |
| 37 | is_valid_palindrome | Checks if a string is a valid palindrome, considering only alphanumeric characters and ignoring case. |
| 38 | is_valid_sudoku | Checks if a 9x9 Sudoku board is valid. |
| 39 | monthly_mortgage_payment | Calculates the monthly mortgage payment based on the loan amount, annual interest rate, and loan term. |
| 40 | note_duration | Calculates the duration between two musical notes based on their frequencies and the tempo. |
| 41 | place_safeway_order | Order specified items from a Safeway location. |
| 42 | polygon_area_shoelace | Calculates the area of a polygon using the shoelace formula. |
| 43 | potential_energy | Calculates the electrostatic potential energy given the charge and voltage. |
| 44 | project_population | Projects the population size after a specified number of years. |
| 45 | reverse_string | Reverses the characters in a string. |
| 46 | solve_quadratic | Computes the roots of a quadratic equation given its coefficients. |
| 47 | trapezoidal_integration | Calculates the definite integral of a function using the trapezoidal rule. |
| 48 | whois | Fetch the WhoIS lookup data for a given domain using the specified Toolbench RapidAPI key. |
| 49 | whole_foods_order | Places an order at Whole Foods. |
| 50 | wire_resistance | Calculates the resistance of a wire based on its length, cross-sectional area, and material resistivity. |

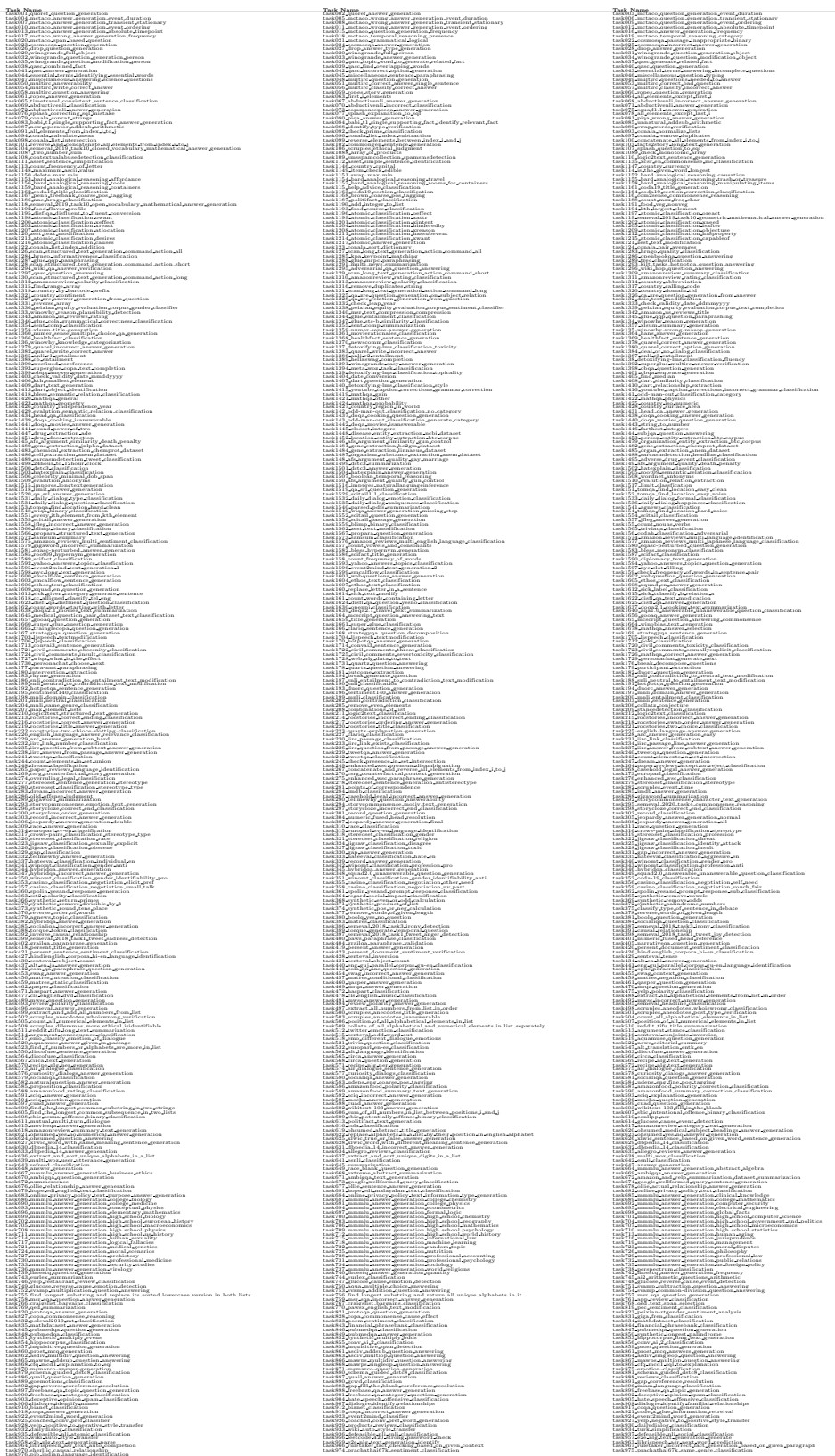

Figure 8: Overview of the 1,000 English tasks from the SNI dataset used in the atomic recall setting.

