# OpenReview forum: "TokMem: One-Token Procedural Memory for Large Language Models"
_ICLR.cc/2026/Conference — ICLR 2026 Poster_

### Official Review · Reviewer_BfBf · 2025-10-26

**Soundness:** 3
**Presentation:** 1
**Contribution:** 2
**Rating:** 4
**Confidence:** 3

**Summary:**

The paper studies the problem of efficiently storing and applying frequently-used procedures (such as calling certain tools) in large language models as opposed to using lengthy prompts or retrieval mechanisms. Specifically, the paper proposes TokMem, which encodes procedural knowledge as trainable continuous memory tokens. The paper evaluates on two settings: using TokMem for single tasks in Super-Natural Instructions and compositionally invoking on function-calling tasks. The results suggest that TokMem outperforms retrieval-augmented generation and parameter-efficient fine-tuning while using fewer parameters.

**Strengths:**

The experimental setting covers multiple models across different scales. The experiment also covers both single task setting and compositional setting.

The results demonstrate good empirical results of TokMem, showing its improvements over LoRA finetuning.

The proposed approach is carefully designed. For instance, TokMem uses the renormalization technique to prevent new tokens from dominating routing.

**Weaknesses:**

I feel the paper needs substantial improvements in terms of clarity.
* The paper heavily uses the term "procedure", but does not give a precise definition of "procedure."
* In particular, it's unclear whether the response in line 129 represents a fixed set for each procedure or can vary.
* The implementation details of memory tokens are ambiguous. Are they essentially implemented as newly introduced special tokens added to the vocabulary?
* I am also not so sure about the routing mechanism. Does the model predict a newly added special token that indexes a procedure during inference?
* DC (decoupled embeddings) seems to also play an important role in some results but it is mainly covered in Appendix.

The choice of baselines mainly uses LoRA finetuning. It would be beneficial to include full fine-tuning as a reference to discuss the trade-offs between the number of parameters updated and performance.

The paper misses discussion of important related work on prompt compression. Methods like Gisting (Mu et al., 2023) and AutoCompressor (Chevalier et al., 2023) that compress prompts into tokens share similar motivations and should be

I'm curious whether new special tokens are necessary. What if we used short natural language descriptions with special token wrappers like <call>tool name</call>, but similarly only tuned representations of these tokens? This approach wouldn't require adding new embeddings for every new procedure.

**Questions:**

See weakness

---

> ### Author Response · Authors · 2025-11-15
>
> We appreciate the reviewer for their recognition of the careful design of TokMem and comprehensive scope of our experiments.
>
> > W1: “The paper heavily uses the term "procedure", but does not give a precise definition of "procedure."”
>
> Thanks for the question. In our work, a procedure is a reusable, context-response mapping that encodes task-specific behavior.
> We used the terminology inspired by psychological research (Anderson & Lebiere, 1998), where procedural knowledge (knowing how, e.g., riding a bike) is distinguished from factual knowledge (knowing that, e.g., it’s a bike).
>
> We’ve added this definition in the introduction.
>
> > W2: “It's unclear whether the response in line 129 represents a fixed set for each procedure or can vary.”
>
> The response for a procedure is not fixed. Each memory token is trained over many instances with varied queries and responses rather than memorizing a single sequence. We have clarified this point in the Section 2.2, where we specify the case for a training instance and a training corpus.
>
> > W3: “The implementation details of memory tokens are ambiguous. Are they essentially implemented as newly introduced special tokens added to the vocabulary?”
>
> Yes. The memory tokens are implemented as newly introduced special tokens added to the model’s vocabulary. This was described in Section 3.1 (“Training Details”). We’ve added additional clarification in Section 2.2.
>
> > W4: “I am also not so sure about the routing mechanism. Does the model predict a newly added special token that indexes a procedure during inference?”
>
> Yes. At inference time the model predicts one of the memory tokens using next-token prediction conditioned on a query. The selected memory token is then appended to the query and generates response autoregressively. If the query is from a compositional task, the model may generate another memory token after each output segment.
>
> We have clarified this process by adding a new section 2.3.
>
> > W5: “DC (decoupled embeddings) seems to also play an important role in some results but it is mainly covered in Appendix.”
>
> Our main method is TokMem without DC (decoupled embeddings) because our experiments show that the extended capacity provided by DC does not improve performance for larger models (Table 1).
>
> We included DC, as it is a tempting variant for analysis. We clarified our intention in Sec 3.2 (Page 6).
>
> > W6: “The choice of baselines mainly uses LoRA finetuning. It would be beneficial to include full fine-tuning as a reference to discuss the trade-offs between the number of parameters updated and performance.”
>
> Thanks for the suggestion. While full finetuning is beyond our computational resources, we have tried increasing training parameters by applying LoRA on all linear layers in our preliminary experiment. We found no improvement but introduced much higher computational cost. For example, LoRA fine-tuning all linear weights of Llama 3B model on 10 SNI task mixture achieves a Rouge-L of 66.8, which is lower than the 67.1 reported for our LoRA baseline in Table 1.
>
> We have clarified this design choice in Footnote 2 for the fine-tuning baseline.
>
> > W7: “The paper misses discussion of important related work on prompt compression”
>
> We thank the reviewer for highlighting Gisting and AutoCompressor. While these methods compress prompts into fixed prefixes, TokMem can recall and compose compressed tokens when needed at generation. We have added a discussion for these two papers in our Section 4. Thanks!

---

> > ### Author Response · Authors · 2025-11-15
> >
> > > W8: “I'm curious whether new special tokens are necessary. What if we used short natural language descriptions with special token wrappers like [object Object]tool name[object Object], but similarly only tuned representations of these tokens?”
> >
> > Thank you for this interesting suggestion. We believe that scaling memory tokens linearly is necessary for accurate memory routing. In the suggested setup, the left wrapper token [object Object] is learned to map a query to a correct tool by generating the tool text. We note that our LoRA finetuning baseline already implements text-based routing by generating tool names before responses, and its tool selection performance as well as the final performance is worse than TokMem as seen in Table 3. This aligns with our intuition that procedural knowledge is nuanced and may not be fully captured by a few words. Thus, we introduced the special memory tokens.
> >
> > The reviewer’s proposed approach would face the same limitation since it requires generating tool names for routing, and in fact it would likely perform worse than LoRA baselines because it just tunes the left wrapper token embeddings instead of the model itself, this single-embedding mapping would be a huge bottleneck for routing, and the error would be propagating to the response generation.
> >
> > > Summary
> >
> > We sincerely thank the reviewer for their feedback which has significantly improved the paper's clarity. We hope the reviewer finds these updates sufficient to support our paper. We would be happy to address any remaining concerns during the discussion period.

---

> > > ### Author Response · Authors · 2025-11-26
> > >
> > > We appreciate your patience as we conducted additional experiments to address your intriguing question regarding the necessity of unique memory tokens (“W8” in our previous response)
> > >
> > > We call this method “wrapper” that learns two special tokens wrapping the textual memory id ([object Object]memory id[object Object]). We implemented it under the setup of Section 3.3 and Table3 for compositional memory recall. Since it is also an embedding-level adaptation method like TokMem, we add an adaptation phase where we first train wrapper embeddings and base model jointly on a held-out dataset, then we freeze the model and re-initialize the wrapper embeddings to avoid the interference effect. We have the following results.
> > >
> > > | Model | Method | Tool Selection (avg) | Argument (avg) |
> > > |:-------|--------:|:------------:|:------:|
> > > | Llama 1B | Wrapper | 59.2 | 41.4 |
> > > |  | + adapt | 68.7 | 47.0 |
> > > |  | TokMem | **98.4** | **85.5** |
> > > | Llama 3B | Wrapper | 70.0 | 56.8 |
> > > |  | + adapt | 80.3 | 63.9 |
> > > |  | TokMem | **99.2** | **86.3** |
> > > | Llama 8B | Wrapper | 76.6 | 61.3 |
> > > |  | + adapt | 90.1 | 79.4 |
> > > |  | TokMem | **99.1** | **89.3** |
> > >
> > > We see that this “wrapper” method cannot outperform TokMem with and without adaptation. This is because relying on just two learnable embeddings creates a bottleneck when they are shared no matter the size of independent memories.
> > >
> > > We hope these additional experiments fully address your curiosity and show the validity of TokMem's design. We are looking forward to your support of our paper. Thank you!

---

### Official Review · Reviewer_unGL · 2025-10-26

**Soundness:** 3
**Presentation:** 3
**Contribution:** 3
**Rating:** 6
**Confidence:** 3

**Summary:**

This paper proposes TokMem, a tokenized memory that stores recurring procedures as compact, trainable embeddings. In particular, frequently reused procedures are “compressed” and stored by encoding them into an internalized memory token. To support continual adaptation, TokMem keeps the backbone model frozen, allowing new procedures to be added without interfering with existing ones.

**Strengths:**

- The paper is generall well-written and easy to follow.

- The proposed method is compared to multiple valid baselines, with complementary experiments to understand the design choices.

- Memory tokenization naturally reduces context size and supports continual learning without interfere

- The backbone is frozen during the training, offering effciency and avoiding catastrophic forgetting.

**Weaknesses:**

The method requires an additional adapation stage for compositional tasks, which seems to break the 'frozen backbone' claim? Also, is it fair to compare 'TokMem + adapt' to fine-tuning? While it is true that the stored procedures are modular via independent memory tokens, the comparison of TokMem without adaptation to fine-tuning in Table 3 seems to suggest limited capability of composing the modular procedures.

**Questions:**

1. Is TokMem+DC using two tokens for each procedure/task? If yes, is it a fair comparison to TokMem? And can you further extend to more tokens?

2. Why is renormalization needed, what is the intuition behind? I see it's helpful empirically but not sure if I understand the reason at Line 150-151. In particular, why new embeddings would develop inflated norms?

3. Clarity: I found the following points confusing to me, requiring more time to catch the ideas.

    - At Line 214-215, it is mentioned that both RAG and TokMem are with 'explicit' memory routing. IMO TokMem is doing 'implicit' routing since the memory token is chosen via next-token prediction, unlike RAG or MoE who has dedicated routing mechanism/architectural component. Can you clarify?
    - Are the training examples formulated by inserting memory tokens between query and response of the original datasets?

---

> ### Author Response · Authors · 2025-11-15
>
> We thank the reviewer for mentioning that our paper is generally “well-written and easy to follow”. We also thank the reviewer for recognizing our comprehensive experiments on TokMem’s design.
>
> > W1: “The method requires an additional adapation stage for compositional tasks, which seems to break the 'frozen backbone' claim”
>
> Thanks for raising the point. Our adaptation stages uses auxiliary tasks that are **different** from the evaluated continual learning tasks. During the continual learning phrase, the base model is indeed frozen, so it does not break the frozen backbone claim. We included such a one-time adaptation stage in the compositional tasks, where the model learns how to interleave responses with memory tokens. Even without such adaptation, our method still outperforms competing baselines (Table 3), showing that our proposed TokMem itself is effective.
>
> We’ve provided more discussion in Sec 3.3.
>
> > Q1: “Is TokMem+DC using two tokens for each procedure/task? If yes, is it a fair comparison to TokMem? And can you further extend to more tokens?”
>
> Yes, TokMem+DC indeed uses two tokens for each procedure/task as shown in Figure 5. Here, our intention is not to advocate for the TokMem+DC method, but to show that such extended capacity does not introduce better performance especially for larger backbone models (Table 1).
>
> It is possible to extend to more tokens in a different way, for example, use 1 token for routing and 2 tokens for conditional generation. But this significantly complicates the training and inference, and may lead to no benefit in performance.
>
> Overall, we advocate for the simple yet effective TokMem (without DC). We clarified our intention in Sec 3.2 (Page 6).
>
> > Q2: “Why is renormalization needed, what is the intuition behind?...why new embeddings would develop inflated norms”
>
> Newer embeddings tend to develop larger norms mainly because the backbone is frozen, the easiest way for a new memory token to become distinguishable is to increase its embedding magnitude. When learning new memory tokens, older ones are frozen, renormalization keeps all memory-token embeddings on a similar scale. Otherwise newer memory develops a slightly larger norm, after softmax, it may dominate the others.
>
> We have added a Figure 7 in the appendix to visualize this phenomenon.
>
> > Q3: “TokMem is doing 'implicit' routing since the memory token is chosen via next-token prediction, unlike RAG or MoE who has dedicated routing mechanism/architectural component. Can you clarify?”
>
> We said that TokMem has an “explicit” routing mechanism, because the memory token is indeed predicted explicitly just like RAG or MoE. The difference is that we incorporate the routing into autoregressive generation, while RAG and MoE train a separate predictor. Moreover, our memory tokens and the tasks (procedural knowledge) have one-to-one correspondence.
>
> To avoid any confusion, we have rephrased “explicit routing” to just “routing” in our revision.
>
> > Q3: “Are the training examples formulated by inserting memory tokens between query and response of the original datasets?”
>
> Yes, in the atomic memory setup, the memory token is inserted between the query and response. For compositional tasks, there are multiple responses, which are interleaved with memory tokens. The layout is shown in Eqn. (2).
>
> > Summary
>
> We thank the reviewer again for recognizing our method and experiments. We have provided point-by-point responses to all the comments raised by the reviewer, and we have also updated our manuscript accordingly. We’re looking forward to your stronger support! Please feel free to let us know if you have further questions.

---

> > ### Comment · Reviewer_unGL · 2025-11-23
> >
> > Thanks for the response. I still have some concerns about the setup and interpretation of results in Table 3.
> > - In Table 1, you compare TokMem directly with Fine-tuning. This looks good.
> > - But in Table 3, you additionally augment TokMem with an adaptation phase (but still call it TokMem), then you compare it with Fine-tuning (which is not augmented). Is it a fair comparison? To me, the TokMem starts from a better 'base model' than Fine-tuning. Or am I missing something?
> > - Therefore, the head-to-head comparison analogous to Table 1 should be *TokMem w/o adaptation vs Fine-tuning*, right?

---

> > > ### Author Response · Authors · 2025-11-24
> > >
> > > Thanks for raising the concern regarding the fairness of the TokMem w/ adaptation vs Fine-tuning w/o adaptation. We actually had tested it in our preliminary study, and found that our “finetuning” baseline cannot benefit from the same adaptation phase as TokMem.
> > >
> > > Specifically, we first finetuned the model with the auxiliary dataset with held-out tools, and then we finetuned it on the target tools for evaluation. We show the results as follows.
> > >
> > > |Model | config | 2 calls | 3 calls | 4 calls | avg |
> > > |:-------------|:--------------|:--------------|:--------------:|:--------------:|:--------------:|
> > > |Llama 1B |  original | 77.3 | 72.6 | 55.8 | 68.6 |
> > > |                 | + adapt |  72.8 | 54.6 | 42.1 | 56.5 |
> > > |                 | + adapt & all linear |  74.1 | 66.7 | 68.4 | **69.7** |
> > > |Llama 3B | original | 87.9 | 86.6 | 82.9 | **85.8** |
> > > |                 | + adapt | 78.9 | 69.5 | 63.2 | 70.5 |
> > > |                 | + adapt & all linear | 79.7 | 72.3 | 89.5 | 80.5 |
> > > |Llama 8B  | original | 87.7 | 86.8 | 88.2 | **87.6** |
> > > |                  | + adapt | 77.6 | 66.3 | 84.2 | 76.0 |
> > > |                  | + adapt & all linear | 84.5 | 78.3 | 89.5 | 84.1 |
> > >
> > > We see that the performance degrades compared with the original standard finetuning. Because it requires the model to learn additional memories of tool sets (with different functions and parameters) that cause interference. Although we can alleviate this interference by finetuning all linear layers, we see it still cannot outperform our original setup.
> > >
> > > By contrast, TokMem avoids interference by abandoning the memory tokens learned for those auxiliary tools (Line 9 at algorithm on Page 16).
> > >
> > > > Summary:
> > >
> > > We thank you again for the question. We did not include these results in our original manuscript because fine-tuning with adaptation underperforms no adaptation. Based on the reviewer’s suggestion, we’ve added an appendix B2 (on page 16)  to include these preliminary results to justify our choice of baselines in the main paper. We hope our response fully addresses your concern.

---

> ### Comment · Reviewer_unGL · 2025-11-27
>
> Thanks for the clarification. I think it makes sense that the 'adapt' can make fine-tuning worse. My point was that, the 'TokMem' (second to last row) in Table 3 should be 'TokMem + adapt', and the last row ('-adapt') is the 'TokMem'. The naming convention is inconsistent with Table 1 in the current form of Table 3. People would get confused because it seems that 'TokMem' contains an 'adapt' procedure or component if looking at Table 3.
>
> Essentially, TokMem needs an initial adaptation phase to outperform fine-tuning for compositional tool-use (unless you consider the adaptation to be a part of it?), and I don't think it's acknowledged in the text (or please let me know if I missed).

---

> > ### Author Response · Authors · 2025-11-27
> >
> > We thank you for highlighting the naming confusion between Table 1 and Table 3. We have revised Table 3 to explicitly label the rows as TokMem (w/ adapt) and TokMem (w/o adapt).  We have also revised the text in Section 3.3 (Lines 376-377) to explicitly define the adaptation phase as a standard component for compositional memory recall. We hope these changes resolve the ambiguity.

---

### Official Review · Reviewer_3EcY · 2025-10-31

**Soundness:** 1
**Presentation:** 1
**Contribution:** 2
**Rating:** 0
**Confidence:** 2

**Summary:**

The paper introduces a method augmenting a language model with a bank of trainable “procedure” embeddings. The model is trained on sequences of query tokens with interleaved "procedure" embeddings and response tokens.

The method is evaluated on "Atomic Memory Recall" and "Compositional Memory Recall" setups.

**Strengths:**

At present I’m unable to identify clear strengths because the core method and its usage at inference are unclear. As a result I can’t assess the contribution or empirical value with confidence.

Conceptually, representing procedures as tokens could be interesting.

**Weaknesses:**

The paper is highly unclear and have multiple problems, for example:
1. The inference process is not described at all. During training, the model learns to predict a sequence of [query tokens]+interleaved [memory token][procedure tokens] -- what happens during inference? Are we appending procedure tokens to the input? In such case, it contradicts the motivation where the method is presented in opposition to e.g. RAG methods that inflate the context window. If the method is not injecting textual tokens of the "procedures", then it's another PEFT method, and should be compared to different PEFT baselines, including prompt-tuning, as well as other dynamic adapter approaches like Mixture-of-LoRAs.
2. As the inference is unclear, so is the evaluation. For the "Atomic Memory Recall" setup, the tasks from Super-Natural Instructions dataset are regarded as the "procedures" in the memory bank. What is exactly evaluated here? Given test query, model predicts the memory token, then we append the text tokens of corresponding "procedure" and evaluate that with Rouge-L?
3. Model is trained on data in the sequential order, which is contrary to the standard of shuffling the data for _stochastic_ gradient descent.
4. Fine-tuning baseline includes only LoRA applied to query & key projections of the attention layers. It should involve also full fine-tuning, or at least LoRA applied to all linear layers, and not an arbitrary subset of them.
5. What was the procedure of selecting hyperparameters for training? For example, learning rate for training LoRAs seems to be too low. Moreover, optimal learning rate usually differs across model sizes -- here the authors provided only a single value.
6. "Routing" is mentioned multiple times in the paper but never introduced or defined.
7. The method is described (e.g. in the abstract) as keeping the backbone frozen. However, in the second evaluation setup the model is initially finetuned before applying the TokMem.

As the authors claim in the Appendix, the paper was written with the help of ChatGPT, which might partially explain the level of presentation of the paper.

After clarifications, I’m open to revising my score.

**Questions:**

All the questions are listed in the weaknesses above.

---

> ### Author Response · Authors · 2025-11-15
>
> We thank the reviewer for taking their time to provide detailed feedback. We hope our clarifications below and also the revision can address the reviewer’s concerns.
>
> > W1: “The inference process is not described at all. During training, the model learns to predict a sequence of [query tokens]+interleaved [memory token][procedure tokens] -- what happens during inference? Are we appending procedure tokens to the input? In such case, it contradicts the motivation where the method is presented in opposition to e.g. RAG methods that inflate the context window. If the method is not injecting textual tokens of the "procedures", then it's another PEFT method, and should be compared to different PEFT baselines, including prompt-tuning, as well as other dynamic adapter approaches like Mixture-of-LoRAs.”
>
> Thanks for the question. TokMem does not add textual procedure descriptions at inference. Instead, the model predicts memory tokens and responses in an autoregressive fashion just like training. We added a subsection on inference (Page 3). Our treatment does **not** contradict the motivation, because we only need one or a few memory tokens which are negligible compared with classic RAG.
>
> Regarding PEFT: In our paper, we have already compared our approach with LoRA in Tables 1 and 3 (called fine-tuning) and prompt-tuning in Tables 4 and 6 .
>
> Our method is not a standard PEFT, as we disentangle different tasks, each learned as a memory token. However, we’re also able to combine multiple tasks by composition, as different memory tokens (and corresponding responses) form an autoregressive chain. This cannot be achieved by traditional PEFT methods such as prompt-tuning and LoRA. Our analysis shows that traditional LoRA suffers from the catastrophic forgetting problem (Figure 3). Even for plug-and-play mixture-of-LoRAs, the mixtures are typically invoked independently (Wenfeng et al. 2024) and are not designed for task composition; we have discussed the difference in related work.
>
> Moreover, our training memory is ~10x lower than LoRA, while achieving higher performance (Table 3). For mixture-of-LoRAs suggested by the reviewer, we assume there are 8 mixtures as in Wenfeng et al. 2024. Its memory use would be 80x larger than ours, so our TokMem clearly has its advantages. We’ve included additional discussion on training parameters on Page 7.
>
> > W2: “As the inference is unclear, so is the evaluation”
>
> We evaluated our method by both routing accuracy and end-to-end task performance.
>
> The routing accuracy measures whether the model predicts the correct memory tokens at inference. Since each sample is tagged with one or more **ground-truth procedure labels**, we can compare the memory token predictions against the ground-truth procedure labels.
>
> For task performance, we disregard the generated memory tokens, but evaluate the generated responses for the task, for example, ROUGE-L for SNI and F1 score for APIGen.
> We’re grateful to the reviewer’s comments. Although we mentioned these metrics in respective experiments (Secs. 3.2 and 3.3), we’ve revised our paper by adding a dedicated paragraph discussing the evaluation methodology (Sec 3.1).
>
> > W3: “Model is trained on data in the sequential order”
>
> We would like to clarify that only tasks are introduced sequentially for the continual learning setup, which mimics the memories that are acquired over time. However, the training samples in each task are shuffled following standard stochastic gradient descent. We have clarified this in the first paragraph of Section 3.2.
>
> > W4: “Fine-tuning baseline includes only LoRA applied to query & key projections of the attention layers.”
>
> Thanks for the catch! Applying to “query & key” was a typo in the paper. We actually fine-tuned the query and value projections, following the original LoRA paper (hu et al. 2022) that recommends this configuration for parameter efficiency. This can be verified in our training scripts in our [anonymized code](https://anonymous.4open.science/r/TokMem-BCBB/) by the arguments “--target_modules "q_proj,v_proj"” . We have corrected this typo in our manuscript.
>
> We also thank the reviewer for the suggestion of applying full fine-tuning or LoRA to all linear layers. We explored this in our preliminary study on 10 SNI tasks and found that increasing training parameters does not bring noticeable improvement and may lead to overfitting because of the limited training samples per procedure.
>
> | Method | Qwen 0.5B  | Llama 3B | Llama 8B |
> |:-------------|:--------------:|:--------------:|:--------------:|
> |   LoRA (q, v)    |   52.4      |   67.1    |   75.8  |
> |   LoRA (q, k, v, o, gate, up, down)   |     53.3      |      66.8     |  75.4   |
>
> We have clarified this design choice by adding a footnote for the fine-tuning baseline in Section 3.1.

---

> > ### Author Response · Authors · 2025-11-15
> >
> > > W5: “What was the procedure of selecting hyperparameters for training?”
> >
> > We select hyperparameters based on the validation performance. For LoRA, we referred to [lora-hyperparameters-guide](https://docs.unsloth.ai/get-started/fine-tuning-llms-guide/lora-hyperparameters-guide) and chose the learning rate from {5e-4, 1e-4, 5e-5, 1e-5, 5e-6} . They perform similarly but 5e-5 gave slightly better performance. For TokMem, the learning rate was chosen from {1e-2, 5e-3, 1e-3} because of the longer gradient path to the input layer, and we finally chose 5e-3 for all experiments.
> >
> > > W6: “"Routing" is mentioned multiple times in the paper but never introduced or defined”
> >
> > Thanks for the comment. Memory routing refers to the selection of the appropriate memory tokens for a given query. We adopted this terminology from other literature, such as mixture-of-expert routing. We’ve now defined routing explicitly in the added Section 2.3 for TokMem inference.
> >
> >
> > > W7: “in the second evaluation setup the model is initially finetuned before applying the TokMem.”
> >
> > We thank the reviewer for raising this point. The adaptation phase is a one-time initialization on auxiliary data teaching the backbone model to compose discrete memory tokens. After this, the backbone remains frozen for memory acquisitions. In Table 3, we show that even without adaptation, TokMem still significantly outperforms ICL and RAG baselines. After adaptation, it better supports continual learning without backbone update, which is different from LoRA that must update the backbone’s weight for memory acquisitions and suffers catastrophic forgetting (Figure 3).
> >
> > > Clarification on writing “the paper was written with the help of ChatGPT, which might partially explain the level of presentation of the paper.”
> >
> > We used LLMs for checking grammar and polishing our paper in a way similar to a phrase/sentence-level thesaurus. All contents were written and verified by the authors.
> >
> > >Summary
> >
> > We thank the reviewer again for the detailed feedback. We’ve addressed all the concerns by the above point-by-point responses and thorough revision of our manuscript. We’re looking forward to your support of our paper.

---

> ### Comment · Reviewer_3EcY · 2025-11-21
>
> Thank you for your response and clarifications. I have a few questions/comments:
>
> 1. After clarifications, this method looks very similar to "Memory Tokens: Large Language Models Can Generate Reversible Sentence Embeddings" by Ignacio Sastre & Aiala Rosá. Could the authors highlight the novelty and differences over the method in the mentioned paper?
>
> 2. Could the authors demostrate a few (not cherry-picked) samples from TokMem and LoRA baseline for the evaluation setups used?
>
> 3. The authors claim that only additional embeddings are trainable (besides small SFT of full model for one experimental setting). What about LM head, the layer outputting distribution over vocabulary? Is it trainable? If not, how does the model learn to predict the new embeddings?
>
> 4. >Our analysis shows that traditional LoRA suffers from the catastrophic forgetting problem (Figure 3)
>
> If the model was trained in a sequential order -- this might be the source of the issue. For _baselines_, I would recommend to train the model in a standard way -- shuffling all samples (across tasks), and decaying LR with small warmup.
>
> 5. Regarding LoRA baseline and configuration -- applying adapters only to attention components is known to be underperforming. For strong baseline, I would recommend applying LoRA to all linear layers and decreasing the rank if needed. As for overfitting -- as mentioned in the previous point -- it's most likely due to sequential training. Training over shuffled dataset of all tasks and samples (and potentially decreasing the rank) should fix the problem.
>
> 6. Regarding hyperparameters -- the learning rate used for LoRA baseline is almost certaintly too low, at least for smaller models. I'd suggest fixing LoRA's alpha parameter (scaling the gradient magnitude) to the used rank, and perform a sweep over learning rates from range e.g. {3e-3, 1e-3, 3e-4, 1e-4}.
>
>
> All abovementioned suggestions together should lead to a strong LoRA baseline for a fair comparison.

---

> > ### Author Response · Authors · 2025-11-24
> >
> > We appreciate the reviewer for raising additional questions, and we address them below.
> >
> > > “this method looks very similar to "Memory Tokens: Large Language Models Can Generate Reversible Sentence Embeddings" by Ignacio Sastre & Aiala Rosá. ”
> >
> > Thanks for bringing up the paper titled “Memory Tokens: Large Language Models Can Generate Reversible Sentence Embeddings”. We were aware of this line of method that compress a single text into a memory token, and in fact our **Section 3.4 already compares against this exact setup** using the method from [1], which shares the exact same core mechanism “reversible sentence embeddings” as Sastre & Rosá. (We’ve cited Sastre & Rosá as well at line 470 in our revision.)
> >
> > We want to note that there are **fundamental differences between our work and theirs in terms of generalization**. Sastre & Rosá use per-sample optimization to memorize a single piece of text, and it cannot generalize to other texts. Our TokMem performs task-level optimization where we train memory tokens on diverse examples so that it learns abstract procedures and generalizes to new queries. More importantly, they condition on a prefix token to reconstruct a single text, but TokMem generates one or many infix tokens that are interleaved within text tokens and support compositionality.
> >
> > [1] Kuratov, Yurii, et al. "Cramming 1568 tokens into a single vector and back again: Exploring the limits of embedding space capacity." ACL 2025
> >
> > > “Could the authors demostrate a few (not cherry-picked) samples from TokMem and LoRA baseline for the evaluation setups used?”
> >
> > We show the generation process of TokMem and LoRA baseline using a query which requires calling a tool “project_population” with id 43 and a tool “investment_profit” with id 29 at test time.
> >
> > Query: What will be the population of a city in 10 years if the current population is 100,000 and the annual growth rate is 2%? Also, calculate the profit from an investment of $2500 at a 4.5% annual rate over 7 years.
> >
> > - What LoRA baseline generates autoregressively:
> >
> > \n[tool_43]{“current_pop": 100000, "num_years": 10, "annual_growth": 2}\n[tool_29]{"amount": 2500, "rate": 0.045, "years": 7}<|eot_id|>
> >
> > where “[tool_43]” and “[tool_43]” are sequences of text tokens with length of 5. We use regex to extract the tool name for routing accuracy and compute F1 on the json arguments.
> >
> > - What TokMem generates autoregressively:
> >
> > \n<|reserved_special_token_43|>{“current_pop": 100000, "num_years": 10, "annual_growth": 2}\n<|reserved_special_token_29|>{"amount": 2500, "rate": 0.045, "years": 7}<|eot_id|>
> >
> > TokMem only needs to generate a single token representing that tool. For evaluation, we check if the predicted token matches the ground truth token id, and compute F1 on the generated arguments.
> >
> >
> > > “What about LM head, the layer outputting distribution over vocabulary? Is it trainable? If not, how does the model learn to predict the new embeddings?”
> >
> > We want to clarify that the LM head is partly trained and partly fixed. For the natural language tokens in the vocabulary, their weights (sometimes called “unembeddings”) are fixed. For the new memory tokens, their weights/unembeddings are trained. We’ve clarified this in the revision (Lines 150-152).
> >
> > > “If the model was trained in a sequential order -- this might be the source of the issue. For baselines, I would recommend to train the model in a standard way -- shuffling all samples (across tasks), and decaying LR with small warmup.”
> >
> > We want to clarify that, while we advocate for the TokMem’s continual learning ability by training with sequential order of tasks, we also have included the non-continual learning setup for both atomic and compositional recall settings.
> >
> > Specifically, in Figure 2, we show the TokMem’s improvement in low-data regime with a 10-task mixture of SNI tasks (all samples and tasks are shuffled, following the standard setting). In Table 3, all results are reported from a shuffled data with all 50 tools. Learning rate is warmed-up followed by a linear decay for both baselines and our method for fair comparison.

---

> > > ### Author Response · Authors · 2025-11-24
> > >
> > > > “For strong baseline, I would recommend applying LoRA to all linear layers and decreasing the rank if needed. For strong baseline, I would recommend applying LoRA to all linear layers and decreasing the rank if needed.”
> > >
> > > and
> > >
> > > > “Regarding hyperparameters -- the learning rate used for LoRA baseline is almost certaintly too low, at least for smaller models.”
> > >
> > > To address your concern on our LoRA baseline. We now apply the LoRA baseline with decreased rank r=4 (previously r=8) on all linear layers (previously on query/value projections). We reuse the setup in Table 3 where we train on a training set of 50 tools with shuffled samples. And we report the argument generation F1 score for each learning rate from {3e-3, 1e-3, 3e-4, 1e-4}.
> > >
> > >
> > > |Model | config | lr | 2 calls | 3 calls | 4 calls | avg |
> > > |:-------------|:--------------|:--------------|:--------------:|:--------------:|:--------------:|:--------------:|
> > > |Llama 1B  | all linear | 3e-3  | 0.0 | 0.0 | 0.0 | 0.0 |
> > > |  | all linear  | 1e-3    | 0.0 | 0.0 | 0.0 | 0.0 |
> > > |  | all linear  | 3e-4    | 77.2 | 71.5 | 63.2 | 70.6 |
> > > |  | all linear  | 1e-4    | 76.7 | 67.1 | 68.4 | 70.7 |
> > > |  | original (q and v) | 5e-5 | 77.3 | 72.6 | 55.8 | 68.6 |
> > > |  | TokMem | 5e-3 | 84.3 |84.3 | 87.8 | **85.5** |
> > > |Llama 3B  | all linear | 3e-3    | 0.0 | 0.0 | 0.0 | 0.0 |
> > > |  | all linear | 1e-3    | 0.0 | 0.0 | 0.0 | 0.0 |
> > > | | all linear | 3e-4    | 81.0 | 71.9 | 84.2 | 79.0 |
> > > | | all linear | 1e-4    | 80.2 | 73.5 | 84.2 | 79.3 |
> > > | | original (q and v) | 5e-5 | 87.9 | 86.6 | 82.9 | 85.8 |
> > > |  | TokMem | 5e-3 | 85.9 | 86.7 | 88.3 | **86.3** |
> > > |Llama 8B  | all linear | 3e-3    | 0.0 | 0.0 | 0.0 | 0.0 |
> > > |  | all linear | 1e-3    | 0.0 | 0.0 | 0.0 | 0.0 |
> > > |  | all linear | 3e-4    | 81.5 | 70.7 | 89.5 | 80.6 |
> > > |  | all linear | 1e-4    | 81.9 | 75.1 | 89.5 | 82.2 |
> > > |  | original (q and v) | 5e-5    | 87.7 | 86.8 | 88.2 | 87.6 |
> > > |  | TokMem  | 5e-3 | 88.1| 86.5 | 93.4 | **89.3** |
> > >
> > > These results show that LoRA struggles at training with higher learning rates, and our original baseline was reasonable and competitive in our experiment. Our learning rate is within the recommended range of the [LoRa Hyperparameters Guide](https://docs.unsloth.ai/get-started/fine-tuning-llms-guide/lora-hyperparameters-guide).
> > >
> > > > Summary:
> > >
> > > We hope our responses address all the reviewer’s additional concerns, and we're looking for your support for our paper this time. Thank you!

---

> > > > ### Comment · Reviewer_3EcY · 2025-11-24
> > > >
> > > > Thank you for your response and additional baseline results. I will raise my score to 6.

---

> > > > > ### Author Response · Authors · 2025-11-24
> > > > >
> > > > > We appreciate your decision to raise the score. We are glad that our response and results have addressed your concerns.

---

### Official Review · Reviewer_9FXM · 2025-10-31

**Soundness:** 3
**Presentation:** 2
**Contribution:** 3
**Rating:** 6
**Confidence:** 3

**Summary:**

This paper introduces TokMem, a trainable memory module that can augment existing language models and steer their behavior. The memory module is represented as a bank of memory tokens that can be retrieved and appended to the context on the fly, and each memory token is trained to be associated with a response. Only the memory embeddings are updated during training while the base model remains frozen.
The experiments include popular approaches, such as ICL, RAG,  FT, and reply memory, and test them on Super Natural Instructions and APIGen, representing atomic and compositional memory.

**Strengths:**

The paper presents a novel approach to managing memory for language models by fine-tuning just a memory module. The results are solid with comprehensive comparisons with different baselines. The design choices are solid and backed by ablation studies. It’s also great that the paper consider different types of memories like atomic and compositional memory.

**Weaknesses:**

In terms of the experimental settings, the original Super Natural Instruction paper uses unseen tasks for testing, but this paper uses the same tasks for training and testing. As a result, it’s unclear how the TokMem can help existing language models generalize to new tasks. Do you have evaluations on applying TokMem to the seen unseen tasks?

The presentation could benefit from how the memory tokens are used during inference. (see below for some questions on the procedure)

Typo: line 312 “fine-tine” → “fine-tune”

**Questions:**

How are the memory tokens retrieved during inference?

---

> ### Author Response · Authors · 2025-11-15
>
> We thank the reviewer for recognizing that our work has “a novel approach”, that “the results are solid”, and that “the design choices are solid and backed by ablation studies”.
>
> > W1: “it’s unclear how the TokMem can help existing language models generalize to new tasks”
>
> Thanks for raising the concern. We’d clarify that the intended scope of our TokMem is to store reusable procedural memory (e.g., how to perform a given task) into explicit memory tokens. This is different from unseen task generalization, which may rely on the backbone model’s pretrained capabilities instead of the memory module. We evaluate our approach where all tasks are seen, and this setup is shared by all baselines, which ensure a fair and controlled comparison.
>
> We’ve revised our abstract to clearly state the scope of our work.
>
> > W2: “The presentation could benefit from how the memory tokens are used during inference” and “How are the memory tokens retrieved during inference?”
>
> Thank you for the suggestion. We have added Section 2.3 to clarify the inference procedure. At inference, the model predicts memory tokens and responses autoregressively in the same layout as training:
>
> $ \big(q_1, \ldots, q_k,  a_{m_i}, a_{r_{i1}}, a_{r_{i2}}, \ldots, a_{m_{j}}, a_{r_{j1}}, a_{r_{j2}}, \ldots, \ldots\big).$
>
>
> > W3: “Typo: line 312 “fine-tine” → “fine-tune”
>
> We appreciate the reviewer for pointing out the typo. We’ve fixed it (new line number: 370).
>
> > Summary
>
> Thank you again for your feedback on our paper. We hope that our clarification and the added section has addressed all the concerns and significantly strengthened the presentation of our paper.

---

> > ### Comment · Reviewer_9FXM · 2025-11-27
> >
> > Thank you for the reply, I will keep my positive score.

---

### Meta-Review · Area_Chair_fkMB · 2026-01-08

**Summary:**

The paper introduces a trainable memory module, TokMem, to store embeddings for recurring procedures. At inference, these embeddings can be appended to the context to steer generation. Their training approach keeps the base model frozen, and only trains these memory embeddings.

The most pertinent concerns raised by the reviewers were the following:
1. Reviewer 9FXM raised concerns with the experiment setup, particularly that there were no tasks in the test set unseen during training.
2. Both reviewers 3EcY and BfBf raised concerns about the lack of comparisons/discussion of novelty over related works such as "Memory Tokens: Large Language Models Can Generate Reversible Sentence Embeddings", Gisting, etc.
3. Reviewer unGL raised concerns about some of the central claims in the paper, i.e. the claim that the base model is kept frozen. Similar issues were also raised by reviewer 3EcY in follow up discussions.

Apart from these, both 3EcY and BfBf had multiple clarification questions about method and experiment setup of the paper in their initial reviews.

**Reviewer Concerns:**

The authors include and convincingly discuss the novelty of their approach over the past works mentioned by the reviewers. the added discussion in the paper is helpful and strengthens the paper.

Furthermore, the authors contain some additional experiments during the rebuttal period in response to reviewers' questions on LoRA training (adding LoRA adaptors to all linear layers) and the wrapper method proposed by reviewer BfBf as an alternate. This discussion, I believe, does address the majority of the concerns of reviewer 3EcY.

The paper will benefit from some re-writing to address the clarification issues raised by multiple reviewers. Some of the claims need to be re-written or tempered (etc. frozen base model as the LM head is partly trained, this part is confusing even after revisions).

**Reviewer Scores:**

Reviewers 9FXM and unGL would keep their marginally above acceptance scores (6). Reviewer 3EcY increased their score to 6 during discussion (I believe this to be legitimate). BfBf would have continued to be on the fence.

---

### Decision · Program_Chairs · 2026-01-26

Accept (Poster)